# Fecal bacteria transplantation replicates aerobic exercise to reshape the gut microbiota in mice to inhibit high-fat diet-induced atherosclerosis

Jie Men[1][◉]*, Hao Li [1][◉], Chenglong Cui[1][◉], Xuedi Ma[1], Penghong Liu[2], Zhengyang Yu[1], Xueyan Gong[3], Youhao Yao[4], Jieying Ren[2], Chengrui Zhao[1], Binyu Song[1], Kaijiang Yin[1], Jianting Wu[1], Wei Liu[5]

1 Fenyang College of Shanxi Medical University, Fenyang, PR China, 2 First Hospital of Shanxi Medical University, Taiyuan, PR China, 3 Third Hospital of Shanxi Medical University, Taiyuan, PR China, 4 Fifth Hospital of Shanxi Medical University, Taiyuan, PR China, 5 Anhui Agricultural University, Hefei, PR China

◉ These authors contributed equally to this work.
* menjie2020@126.com

**Data Availability Statement:** The Illumina sequencing reads presented in this study have been deposited in the NCBI Sequence Read Archive

## Abstract

Aerobic exercise exerts a significant impact on the gut microbiota imbalance and atherosclerosis induced by a high-fat diet. However, whether fecal microbiota transplantation, based on aerobic exercise, can improve atherosclerosis progression remains unexplored. In this study, we utilized male C57 mice to establish models of aerobic exercise and atherosclerosis, followed by fecal microbiota transplantation(Fig 1a). Firstly, we analyzed the body weight, somatotype, adipocyte area, and aortic HE images of the model mice. Our findings revealed that high-fat diet -induced atherosclerosis mice exhibited elevated lipid accumulation, larger adipocyte area, and more severe atherosclerosis progression. Additionally, we assessed plasma lipid levels, inflammatory factors, and gut microbiota composition in each group of mice. high-fat diet -induced atherosclerosis mice displayed dyslipidemia along with inflammatory responses and reduced gut microbiota diversity as well as abundance of beneficial bacteria. Subsequently performing fecal microbiota transplantation demonstrated that high-fat diet -induced atherosclerosis mice experienced weight loss accompanied by reduced lipid accumulation while normalizing their gut microbiota profile; furthermore it significantly improved blood lipids and inflammation markers thereby exhibiting notable anti-atherosclerosis effects. The findings suggest that aerobic exercise can modify gut microbiota composition and improve high-fat diet-induced atherosclerosis(Fig 1b). Moreover, these beneficial effects can be effectively transmitted through fecal microbiota transplantation, offering a promising therapeutic approach for managing atherosclerosis.

database under the study number PRJNA1106992 and can be accessed at https://www.ncbi.nlm.nih.gov/sra/PRJNA1106992.

**Funding:** This study was supported by the National Natural Science Foundation of China (82201691); Cultivation Key laboratory of Metabolic Cardiovascular Diseases Research (TSGJ001); the Science and Technology Innovation Program of the Shanxi Provincial Department of Education (2023L472); the Science and Technology Plan Project of Lvliang City (2023SHFZ48); and the Talent Introduction Startup Fund of Fenyang College, Shanxi Medical University (2022A01).

**Competing interests:** The authors have declared that no competing interests exist.

## 1. Introduction

Although drugs have been extensively utilized over the past three decades, cardiovascular disease (CVD) remains the predominant cause of global mortality, accounting for 33% of all deaths worldwide. The number of fatalities has surged from 12.1 million in 1990 to a staggering 20.5 million in 2021, representing an alarming increase of 69.42% [1]. Atherosclerosis (AS), which can lead to debilitating and fatal consequences such as myocardial infarction, stroke, and aortic aneurysm, is primarily responsible for CVD-related morbidity [2]. The gut microbiota (GM) exhibits high plasticity and susceptibility to various factors including diet and exercise [3], with its dysfunction playing a pivotal role in promoting AS initiation and progression. Research has demonstrated that alterations in GM composition are observed among patients with atherosclerosis (AS) compared to healthy individuals [4], and the GM, along with its metabolite Trimethylamine-N-oxide (TMAO), increase with the progression of AS [5]. Moreover, substantial variations in the composition and functionality of GM exist among distinct patient subtypes [6], particularly in the context of high-fat diet (HFD)-induced inflammation that accelerates AS. This finding is closely linked to the augmented abundance of pro-inflammatory GM induced by HFD [7], ultimately heightening the susceptibility to AS. Reshaping GM emerges as a promising avenue for the prevention and amelioration of AS, thereby constituting a potential research trajectory.

Currently, three widely used methods for treating AS involve GM utilization. The first method involves altering GM composition through statin drug administration to reduce lipid levels and improve AS [8]. For example, Zhang et al. discovered that simvastatin inhibits AS by modulating GM diversity, abundance ratio, and downstream metabolites [9]. However, long-term usage may lead to various side effects such as muscle pain, weakness, and liver and kidney damage [10, 11]. The second method involves supplementing probiotics to enhance GM diversity and rectify GM imbalance caused by AS [12], while also strengthening intestinal barrier function [13]. Nevertheless, the current use of probiotics faces significant challenges including obstacles to successful implantation and potential safety concerns post-implantation [3]. The third method involves remodeling GM through aerobic exercise to improve AS. An increasing body of evidence has demonstrated that exercise can ameliorate metabolic abnormalities, elicit anti-inflammatory responses within the organism [14], effectively elevate intestinal short-chain fatty acid (SCFA) levels [15], and modulate GM to mitigate AS induced by HFD. Considering the contraindications and risks of exercise, there are obstacles to promoting exercise therapy for AS among the elderly population [16]; Fecal microbiota transplantation (FMT) offers a novel approach for treating this condition. In addition to these three methods, based on existing research results, utilizing FMT technology for reshaping GM represents another promising therapeutic avenue with great potentiality. This primarily revolves around regulating the production of diverse metabolites by GM while blocking pathways leading to the onset of AS; it possesses characteristics such as rapid efficacy, high efficiency along with low immune rejection response [17]. With the continuous advancement of genomics, the utilization of FMT technology for reshaping GM also offers technical support for future targeted microbiota screening and transplantation [18]. Despite attempts by some scholars to use healthy donor FMT for treating and improving diseases by reshaping GM, a therapeutic approach involving aerobic exercise-based FMT to improve AS in mouse GM has not been proposed or validated yet.

Through the dual reactivation of aerobic exercise reshaping GM and FMT techniques, this study aims to establish aerobic exercise, AS, and FMT mouse models to evaluate whether the improvement of AS through GM reshaping by aerobic exercise can be effectively transferred via FMT. This research provides a theoretical basis and new potential therapeutic strategies for preventing and treating AS.

## 2. Material and methods

### 2.1 Experimental animals and grouping

The study utilized a total of forty 3-week-old male C57 mice with SPF-grade, weighing between 11–13 g, which were obtained from the Animal Center of Shanxi Medical University (license number SCXK(Jin)2019-0004). All animals were housed in a temperature-controlled room (temperature: 25 ± 1˚C; humidity: 55 ± 5%; light/dark cycle: 12 hours), and provided ad libitum access to food and water. Following one week of acclimatization, the mice were randomly assigned into four groups: control group (C group, n = 10), atherosclerosis group (AS group, n = 10), aerobic exercise group (A group, n = 10), and fecal microbiota transplantation group (F group, n = 10). Three mice were excluded due to reasons related to FMT or exercise intervention, resulting in a final sample size of n = 37 including C group with n = 10, AS group with n = 9, A Group with n = 9, and F Group with n = 9. The animal experiments received approval from the Ethics Committee of Shanxi Medical University Fenyang College and strictly adhered to guidelines and regulations for animal experimentation. Ethical review number:2023020.

### 2.2 Animal models and intervention strategies

Animal model: C group and A group were fed with regular diet for 19 weeks; AS group was induced to establish the model by feeding HFD (lipid content of 44%) for 11 weeks, and then randomly selected AS model mice as F group. After the 11th week, F group received an 8-week FMT intervention, which means fresh feces from A group were processed and transplanted into AS group. During this period, both AS and F groups continued to be fed with HFD (Fig 1a). Exercise program: The swimming intervention in A group referred to Taka-aki Okabe's

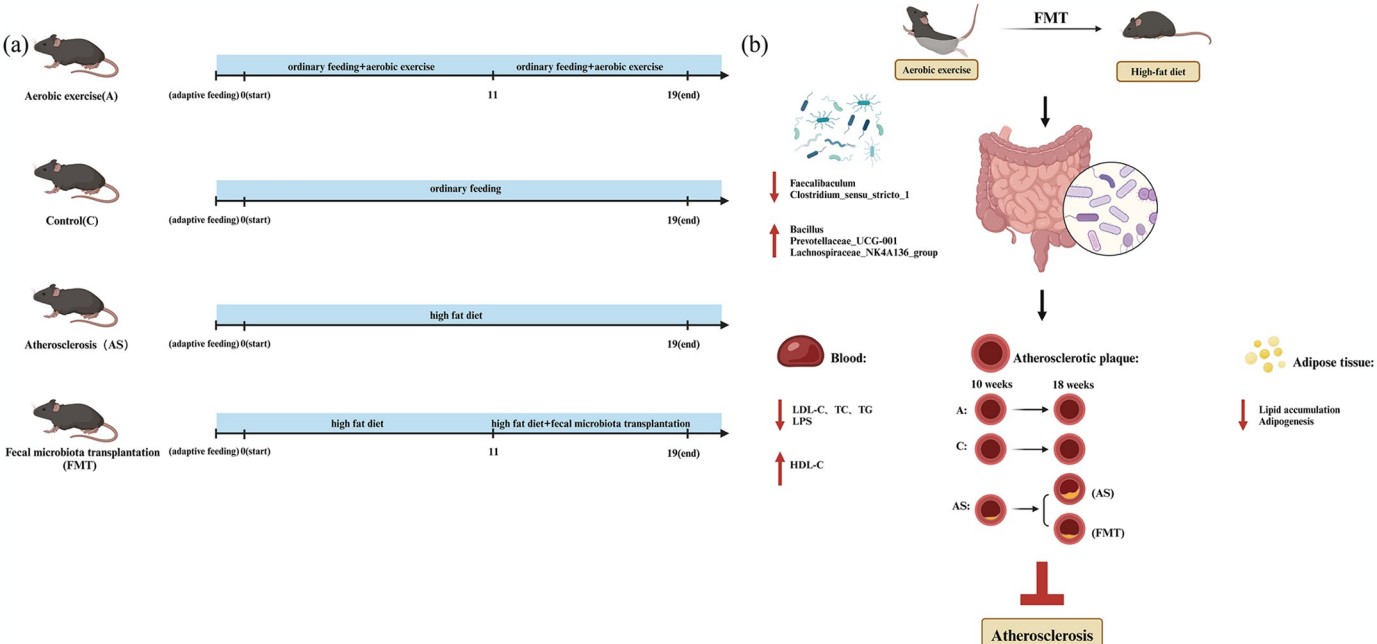

**Fig 1. Fecal bacteria transplantation can replicate the effects of aerobic exercise, resulting in an increase in beneficial bacteria and the inhibition of atherosclerosis.** (S1 Graphical abstract. Created in BioRender. Dongping, L. (2024) BioRender.com/ d59q136). (a)Schematic representation of mouse model development and experimental protocols across groups. (b) Alterations in atherosclerosis-related factors induced by fecal microbiota transplantation in high-fat diet-induced atherosclerotic mice.

protocol with improvements [19]. The exercise dosage was set at 60 minutes per day, 5 days per week for a total of 19 weeks. On the first day, the exercise time was set at 20 minutes and gradually increased until reaching a duration of 60 minutes per day. FMT protocol: The dosage of FMT was set at 0.2ml per administration every other day for a total of eight weeks. Specific operation details regarding FMT can be found in section 2.4, and it should be noted that the experimental periods mentioned above do not include the initial one-week adaptation feeding stage.

## 2.3 Collection of body weight, blood and pathological specimens

During the feeding period, the health status of the mice was observed daily, and body weight was measured weekly at a fixed time to plot the weight trend graph. Mice were fasted for 12 hours and water-deprived for 4 hours before the end of the experiment. All surgical procedures were conducted under pentobarbital anesthesia to minimize the suffering experienced by the mice, and blood, thoracic aorta, abdominal aorta, and adipose were collected. After allowing the blood to stand at room temperature for 30 minutes, it was centrifuged at 3000 rpm for 15 minutes at 4°C. The supernatant serum was transferred to a new EP tube, labeled, and stored in a -20°C freezer. Portions of the thoracic and abdominal aorta and adipose from each group were placed in labeled Microtubes, snap-frozen in liquid nitrogen, and another part was fixed in 4% paraformaldehyde buffer for paraffin embedding.

## 2.4 Fecal microbiota transplantation

Based on our previous experiments that identified GM as a critical factor in disease progression, we conducted FMT experiments for further validation. Freshly excreted feces were collected from Group A (donor mice) during the experiment and promptly frozen. After thawing, feces were made into a 200 mg/mL fecal slurry using phosphate-buffered saline (PBS) under a sterile workbench. The fecal slurry was centrifuged at 1000 rpm for 5 minutes at 4°C, and the supernatant was transferred to new sterile centrifuge tubes, packaged, and frozen at -80°C. Before the FMT procedure, the supernatant was incubated in a 37°C incubator to revive the microbial community until ready for transplantation. Following aliquoting into sterile 1.5 ml centrifuge tubes, the inoculum was administered to the recipient mice in group F via gastric gavage with a dose of 0.2 ml per mouse.

## 2.5 Histological analysis of aorta and white adipose

The mouse aorta and white adipose tissue were fixed in 4% paraformaldehyde buffer. The tissues were dehydrated to wax according to the set program and placed in embedding frames containing melted wax. Once the wax solidified, the wax blocks were removed from the frames, labeled, and stored in a 4°C refrigerator. Tissue sections approximately 4 μm thick were prepared for H&E staining. Images were observed and captured under an optical microscope at appropriate magnifications. White adipocytes size was calculated using ImageJ software (NIH, USA).

## 2.6 Analysis of 16S rRNA sequences

All 16S rRNA amplicons were processed following the QIIME2 (Quantitative Insights into Microbial Ecology 2) [20] pipeline. The DADA2 [21] plugin in this process was used to denoise the optimized sequences and the sequences after DADA2 denoising were defined as ASVs (Amplicon Sequence Variants). Subsequent data analyses were conducted on the Majorbio Cloud platform (https://cloud.majorbio.com). Sample data were processed using mothur

software (http://www.mothur.org/wiki/Calculators, version 1.30.2) to determine α-diversity indices such as Shannon and ACE. Principal Coordinates Analysis (PCoA) using the Bray-Curtis distance algorithm was performed to visually depict the similarity of GM among different sample groups. The LEfSe analysis (Linear Discriminant Analysis Effect Size) [22] (http://huttenhower.sph.harvard.edu/LEfSe) was employed to identify significantly different bacterial taxa at the phylum to genus levels among the mouse groups subjected to different modeling approaches (LDA > 4, $p < 0.05$). Additionally, Spearman correlation analysis was conducted to assess the correlation between the abundance of various bacterial taxa and the concentrations of atherogenic-related blood parameters.

## 2.7 Measurement of serum biochemical parameters

According to the manufacturer's instructions, an automated analyzer was employed for the detection of various biochemical parameters in serum. The measurement of serum triglyceride (TG) and total cholesterol (TC) levels was performed using CheKine™ Triglyceride (TG) and Total Cholesterol (TC) Assay Kits (Micro Method). Additionally, determination of serum high-density lipoprotein cholesterol (HDL-C) and low-density lipoprotein cholesterol (LDL-C) levels was carried out using CheKine™ High-Density Lipoprotein (HDL-C) and Low-Density Lipoprotein (LDL-C) Assay Kits respectively. Specifically, HDL-C measurements were conducted with Abbkine Biotechnology Co., Ltd.'s KTB2250 kit from Wuhan, China while LDL-C measurements utilized Abbkine Biotechnology Co., Ltd.'s KTB2260 kit from the same source. Serum lipopolysaccharide levels were quantified employing LPS obtained from Beyotime Biotechnology Co., Ltd.'s SU-B31044 product based in Wuhan, China.

## 2.8 Statistical analysis

The experimental data were processed and analyzed using GraphPad Prism 9.5 software. If the datasets exhibited a normal distribution and homoscedasticity, Student's t-test and one-way ANOVA were employed; otherwise, the Kruskal-Wallis test was utilized. The study findings were presented as mean ± standard deviation ($\bar{x} \pm SD$), with statistical significance defined as $p < 0.05$.

# 3. Results

## 3.1 Aerobic exercise and FMT improve obesity and AS-related histological changes

To assess the impact of aerobic exercise, HFD, and FMT interventions on AS-related indicators in mice, as well as to investigate whether aerobic exercise-based FMT can modify the phenotype of AS mice, we established an aerobic exercise model and induced an AS model using HFD. Subsequently, the AS group received FMT for 8 weeks. Consistent with previous studies [23], HFD exhibited a significant increase in body weight, body size, visceral adipose mass, adipocyte area, and AS plaque formation (Fig 2), while the A group and F group demonstrated contrasting changes. Compared to the AS group, the F group exhibited a significant decrease in body weight. After FMT intervention, the mice in the F group experienced a rapid initial decrease in body weight within one week, followed by a subsequent stable and gradual increase (Fig 2a). This phenomenon may be attributed to early intragastric administration causing discomfort to the mice and subsequently leading to reduced food intake. Despite an increase in body weight after adaptation to FMT intervention, the overall upward trend remained more moderate compared to that observed in the AS group during the same period ($p < 0.001$). Compared to the AS group, both the A and F groups exhibited significant enhancements in mouse

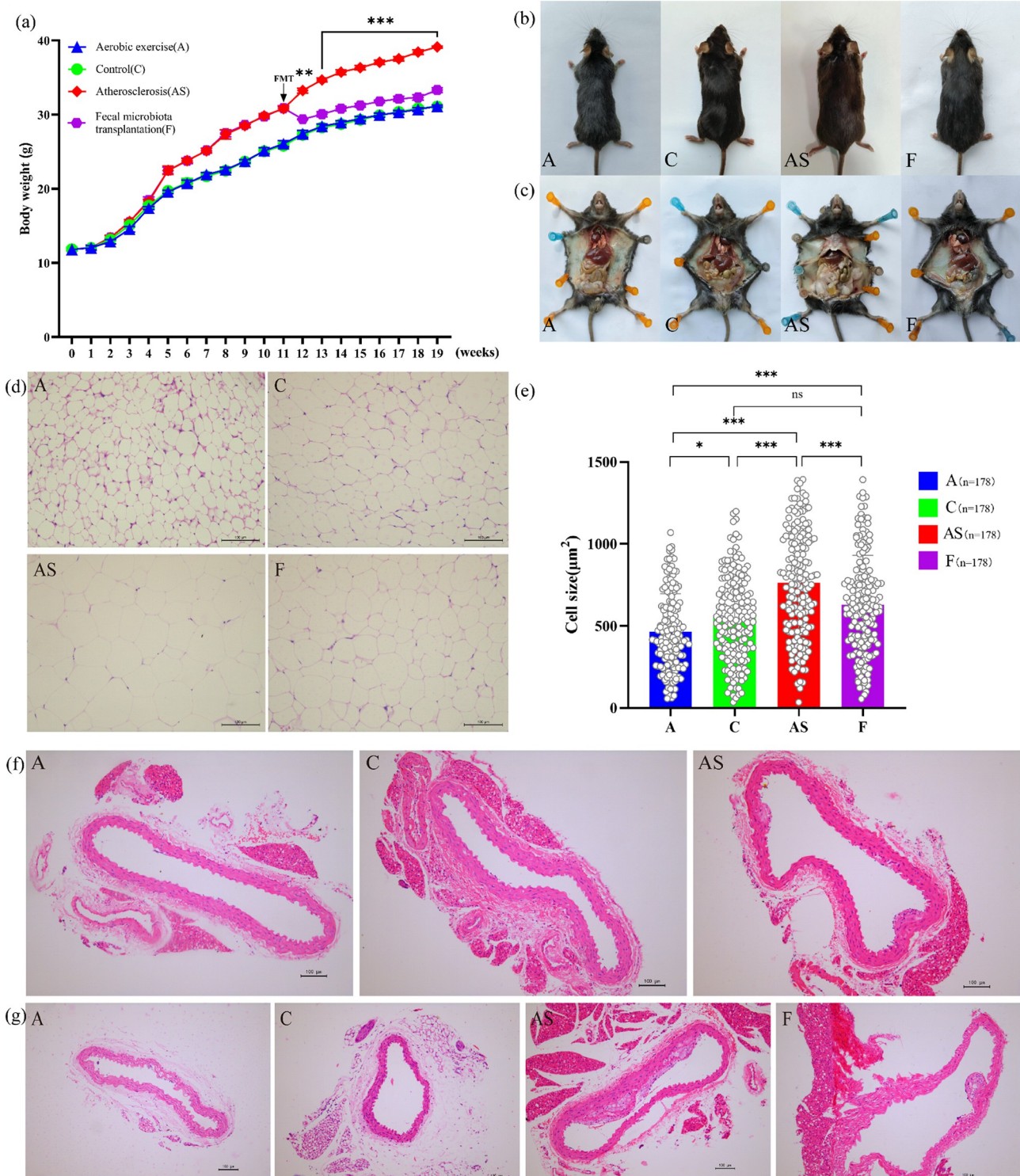

**Fig 2. Aerobic exercise and FMT effectively improve HFD-induced obesity and AS-related histological changes.** (a) Body weight changes throughout the experimental period. (b, c) Representative pictures of the appearance morphology and (c) the amount of visceral adipose in each group of mice. (d) White adipose sections stained with H&E. (e) Mean area of white adipocytes; *$p<0.05$, **$p<0.01$, ***$p<0.001$. (f, g) Pathological state of the aorta stained with H&E (f) after 11 weeks of rearing and (g) after 19 weeks of rearing (scale bar = 100μm). The data presented in Fig 1(a) were analyzed using Student's t-test, revealing a significant difference between the AS and F groups denoted by **$p<0.01$ and ***$p<0.001$. The data in Fig 2(e) were analyzed using Kruskal-Wallis test, revealing the significant differences between groups are indicated by ns $p>0.05$, *$p<0.05$, and ***$p<0.001$.

body size and visceral adipose mass (Fig 2b and 2c) Morphological observations revealed that adipocyte enlargement and plaque development were significantly ameliorated in the A and F groups compared to the AS group (Fig 2d–2g). These findings suggest that aerobic exercise improves lipid accumulation and mitigates AS progression through FMT. Considering that HFD-induced obesity and lipid accumulation are underlying factors contributing to dyslipidemia and inflammatory reactions associated with AS occurrence and progression, FMT should be acknowledged as a potential therapeutic approach for treating AS.

## 3.2 Aerobic exercise and FMT reduce AS risk factors

Insufficient physical activity, hyperlipidemia, and inflammatory response accelerate AS progression [24]. Conversely, regular aerobic exercise exerts a positive impact on ameliorating lipid abnormalities and inflammation. Therefore, investigating whether this favorable alteration can be effectively transferred and replicated in the AS model through FMT represents a novel therapeutic approach. Multiple biochemical and inflammatory markers were assessed and analyzed, revealing significant elevations in LDL-C and TG levels in HFD-induced AS mice compared to the C group ($p < 0.01$, $p < 0.001$; Fig 3a and 3c) Additionally, TC demonstrated a notable elevation ($p < 0.001$; Fig 3b). On the other hand, HDL-C levels in HFD-induced AS mice significantly decreased ($p < 0.001$ Fig 3d), whereas Group A ameliorated lipid abnormalities. Notably, no differential alterations were observed in HDL-C, TC, and TG between Group A and Group F. These findings suggest that aerobic exercise effectively mitigates lipid abnormalities through the efficient migration of FMT, thereby providing further support for the hypothesis of this study. LPS promotes systemic inflammatory responses

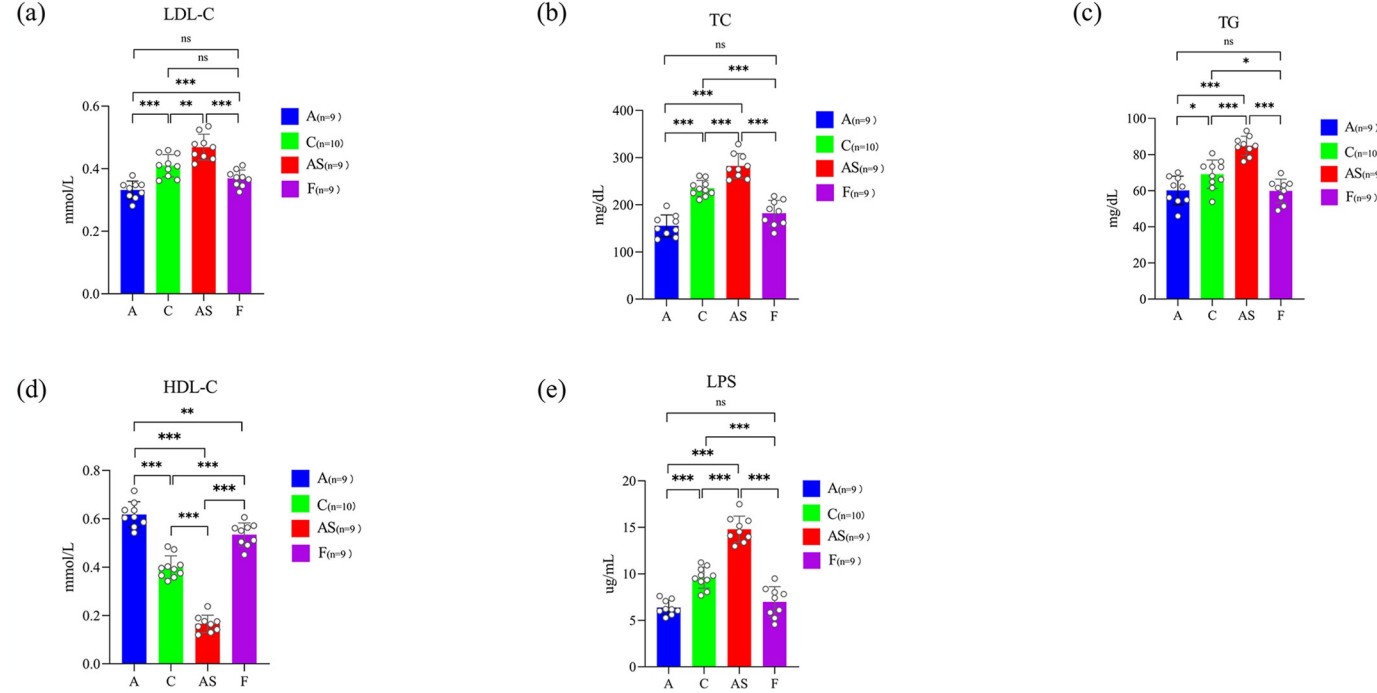

**Fig 3. Aerobic exercise and FMT improve HFD-induced inflammation.** Relative levels of AS-related risk factors in serum. (a) LDL-C; (b) TC; (c) TG; (d) HDL-C; (e) LPS. The data in Fig 3. were analyzed using one-way ANOVA, and the significant differences between groups are indicated by ns $p > 0.05$, *$p < 0.05$, **$p < 0.01$, and ***$p < 0.001$.

through various pathways, disrupts intestinal barrier integrity, accelerates foam cell formation, and its expression level is positively correlated with the severity of AS [25]. Therefore, inhibiting LPS generation is considered an important target for AS treatment. The HFD significantly increased the level of LPS compared to Group C ($p<0.001$). In contrast, Group A showed a significant improvement in this alteration ($p<0.001$; Fig 3e). Furthermore, FMT also replicated the aerobic exercise-induced improvement of the LPS phenotype. The beneficial changes in the phenotype of FMT mice were mainly due to aerobic exercise promoting the production of beneficial bacteria producing short-chain fatty acids (SCFAs), enhancing the integrity of the intestinal mucosal barrier, reducing the proportion of pathogenic bacteria such as gram-negative bacteria, inhibiting LPS escape, and alleviating the body's inflammatory response.

### 3.3 FMT replicates the role of aerobic exercise in improving GM abundance and diversity in HFD-induced AS mice

Aerobic exercise plays a key role in preventing and treating AS by reshaping the GM [26, 27]. However, the assessment of whether aerobic exercise-based FMT remodelled GM in HFD-induced AS mice is unprecedented. To verify whether FMT can replicate the GM phenotype of aerobic exercise to achieve an increase in abundance and diversity of the GM in HFD-induced AS mice. Under the premise that the samples meet the sequencing standards (Fig 4a), we used ACE and Shannon indices to analyze the abundance and diversity of samples in each group. The A group can significantly increase the ACE and Shannon index of mice compared with the AS group, and similar changes were also observed in the F group ($p<0.05$; Fig 4b and 4c). In order to analyze the similarity and difference between samples in each group, we conducted PCoA analysis to visualize the differences in GM structure between the four groups (Fig 4d; PC1 and PC2 account for 10.72% and 8.48% respectively). In the PCoA plot, a clear group-based clustering pattern was observed, with most of the AS group distributed in the lower left quadrant, while representatives of the other three groups were mostly located in the upper quadrants, indicating significant differences in the GM composition between the AS group mice and the other three groups. Additionally, most samples from the A and F groups were concentrated in the upper left quadrant, indicating a high degree of similarity between these two groups. The Venn diagram shows the overlap and exclusivity of core GM in each group (Fig 4e). Among 3717 ASVs, there were 240 overlapping ASVs among the four groups, with 991, 825, 891, and 1010 specific ASVs in groups A, C, AS, and F, respectively. Additionally, the core microbiota shared by the gut of the F group and the AS and A groups were significantly different, with 511 and 568 ASVs, respectively, which is consistent with the diversity results analyzed by the Shannon index. The analysis of GM abundance and diversity results indicates that aerobic exercise can significantly increase GM diversity and abundance. Moreover, the alterations observed in FMT resemble those observed in Group A, providing further evidence that aerobic exercise reshapes the abundance and diversity of GM induced by a HFD, which can be effectively transferred through FMT.

### 3.4 FMT replicates the role of aerobic exercise in reshaping the GM composition of HFD-induced AS mice

It is a consensus that aerobic exercise reshapes the GM, and the GM after FMT is influenced by the donor. The GM of groups A, C, AS, and F was initially analyzed from the phylum level to the genus level. At the phylum level, Firmicutes and Bacteroidetes were dominant, comprising over 85% of the total sequences (Fig 4f). The Firmicutes/Bacteroidetes ratio (F/B) is an important indicator to evaluate GM dysbiosis [28] and is also closely related to aerobic exercise. The F/B ratio was found to be relatively increased in group A compared with group C, and it also exhibited an increase in group F compared with group AS, which is consistent with

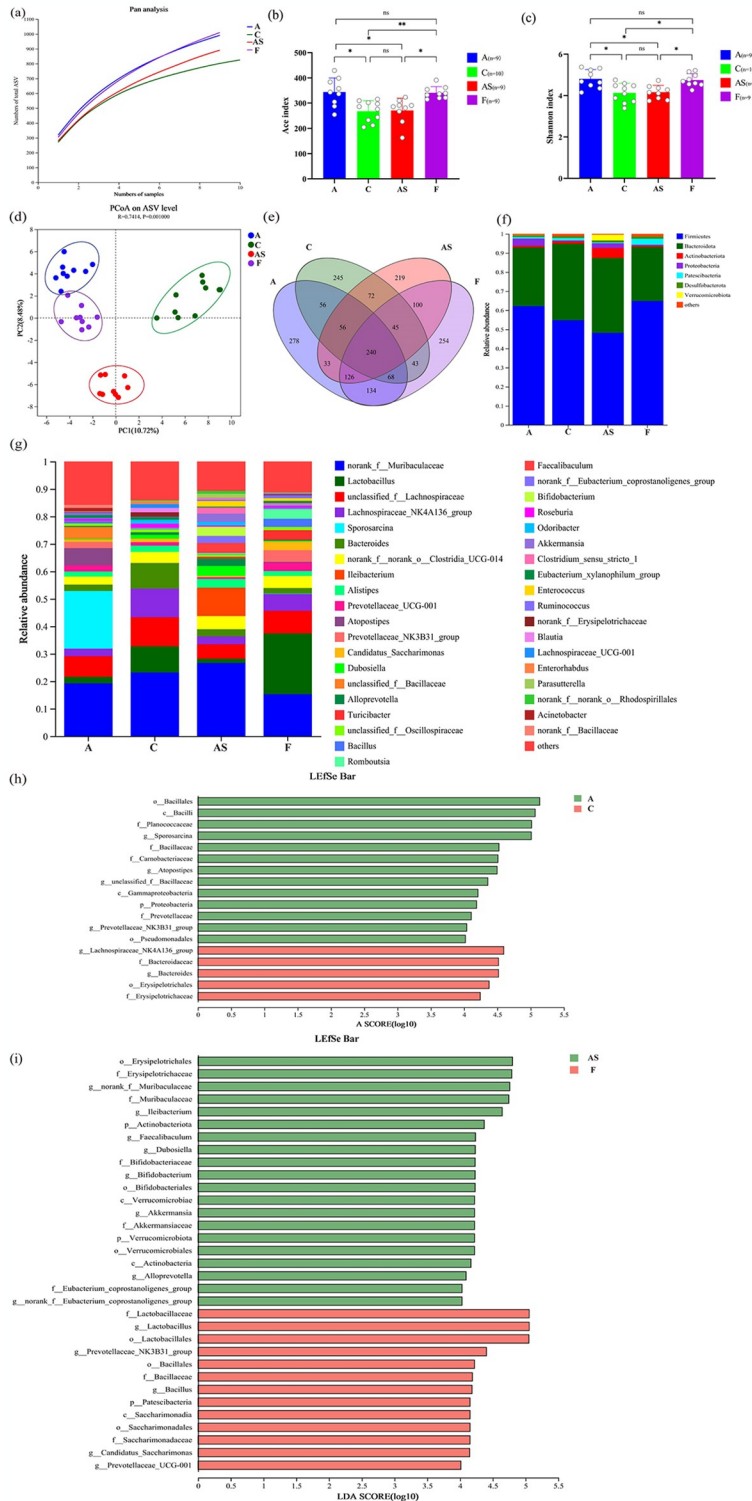

**Fig 4. FMT imitates aerobic exercise's benefits on GM, raising its numbers and transforming its configuration.** (a) Pan species analyses of samples. (b, c) Differences in α-diversity between groups were assessed by (b) shannon index and (c) ace index, respectively. ns $p > 0.05$, * $p < 0.05$, ** $p < 0.01$. (d) Bray-Curtis β-diversity was visualised using principal coordinate analysis (PCoA). (e) Venn diagram of OUT for each group. (f, g) Major GM at (f) phylum level and (g) genus level for each sample (h, i) Based on the LDA score histogram generated by LEfSe analysis, the taxonomic contribution of mouse microbial communities between (h) groups A and C and (i) groups AS and F was

described (LDA > 4.0). The data in Fig 4(b) and 4(c) were analyzed using Kruskal-Wallis test, and the significant differences between groups are indicated by ns $p>0.05$, *$p<0.05$, and **$p<0.01$.

previous findings [29]. Additionally, we found that *Faecalibaculum*, *Clostridium_sensu_stricto_1*, *norank_f_ Erysipelotrichaceae*, and *Parasutterella*, which are significantly enriched through HFD and involved in lipid metabolism, and inflammatory responses, were in higher abundance in the AS group [30], whereas in the A group *Actinobacteriota* and *Verrucomicrobiota*, which are highly associated with CVD risk factors, were reduced in relative abundance. Meanwhile, at the genus level, we also found that FMT remodelled the GM composition of HFD-induced AS mice, including elevated abundance of beneficial bacteria such as *Prevotellaceae_UCG-001*, *Prevotellaceae_NK3B31_group*, *Bacillus*, and *Ruminococcu*, which inhibit AS process, and decreased abundance of harmful bacteria such as *Lachnospiraceae_NK4A136_group*, *Dubosiella*, *noran_f_Erysipelotrichaceae*. The LEfSe discriminant bar chart showed that, compared to group C, the dominant bacteria in group A were c_Bacilli, o_Bacillales, o_Pseudomonadales, f_Planococcaceae, and f_Prevotellaceae. Compared to AS, the dominant bacteria in group F were c_Saccharimonadia, o_Lactobacillales, o_Bacillales, and f_Lactobacillaceae. The dominant bacteria in the AS group were c_Verrucomicrobiae, o_Erysipelotrichales, f_Erysipelotrichaceae, and f_Akkermansiaceae. It is noteworthy that the dominant microbial community in groups A and F primarily comprises beneficial bacteria, demonstrating a high level of similarity. In contrast, group AS mice harbor a predominant microbial community predominantly composed of pathogenic bacteria. These findings suggest successful colonization of group F by group A through FMT, leading to reshaping of the GM composition and further validating the efficacy and feasibility of FMT.

## 3.5 The GM of HFD-induced AS mice is highly correlated with the risk factors of AS

The pathogenesis of HFD-induced AS involves dyslipidemia, heightened inflammation, and imbalances in GM, which aligns with the findings of this study. To further clarify the relationship between GM and AS, as well as provide compelling evidence that modifying relevant bacterial species in GM is one of the underlying mechanisms by which aerobic exercise reduces AS risk factors, we selected the top 40 genera for Spearman correlation analysis at the genus level (Fig 5a). This analysis establishes a strong foundation for FMT studies. The results showed that the relative abundance of most key intestinal beneficial genera, including *Prevotellaceae_UCG-001*, *Prevotellaceae_NK3B31_group*, *Bacillus*, *Candidatus_Saccharimonas*, *Sporosarcina*, and *unclassified_f_Bacillaceae*, was negatively correlated with AS risk factors, and the HDL-C levels are positively correlated with all other genera, except for acetic acid producers and potential probiotic *Bacillus* [31]. In contrast, the relative abundance of harmful genera, including *Clostridium_sensu_stricto_1*, *Dubosiella*, *Faecalibaculum*, and *norank_f_Eubacterium_coprostanoligenes_group*, was positively correlated with AS risk factors and negatively correlated with HDL-C levels. However, in the results, *Blautia* [31], an acetic acid (AA) producer and potential probiotic, showed a positive correlation with LPS levels.

We conducted a two-factor correlation analysis of GM samples (Fig 5b and 5c). The analysis results showed that the order and family level, o_Bacillales, o_Pseudomonadales, f_Planococcaceae, and f_Bacillaceae were negatively correlated with at least two of the AS-promoting LDL-C, TC and LPS, and Erysipelotrichales were positively correlated with LDL-C levels and negatively correlated with HDL-C levels. Combined with the previous results, we observed an

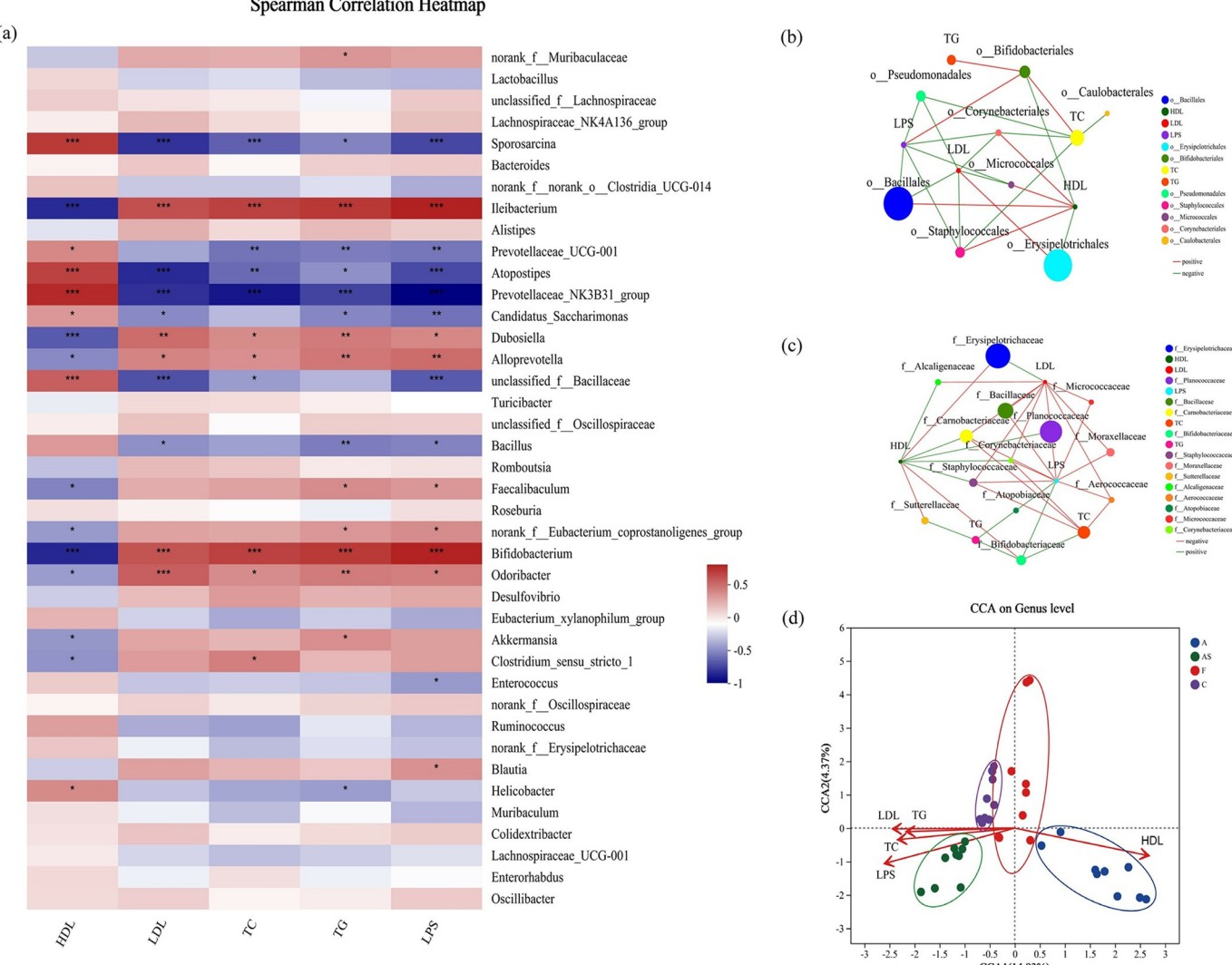

**Fig 5. GM of HFD-induced AS mice is highly correlated with AS risk factors.** (a) Heatmap showing the correlation between selected genera in each group and AS-related indicators. Red color indicates positive correlation and blue color indicates negative correlation. (b, c) Correlation analysis of different bacterial strains with AS risk factors in mice at (b) order level and (c) family level. (d) Correlation analysis between risk factors for AS in mice, samples of each group, and GM. *$p<0.05$, **$p<0.01$, ***$p<0.001$.

interesting phenomenon: beneficial genera such as o_Bacillales, o_Pseudomonadales, f_Plano-coccaceae, and f_Bacillaceae appeared as dominant microbial communities in groups A or F, while harmful genera such as o_Erysipelotrichales and f_Erysipelotrichaceae were the dominant GM in the AS group. This result indicates that aerobic exercise and FMT effectively increase the relative abundance of beneficial bacteria in the host GM. In addition, the results of CCA revealed that in group A, GM exhibited a positive correlation with HDL-C levels and a negative correlation with other AS risk factors. Conversely, the AS group induced by HFD displayed contrasting outcomes (Fig 5d).

## 4. Discussion

In this study, aerobic exercise modulates the composition of GM and can be effectively transferred through FMT to ameliorate dyslipidemia and inflammation in AS. The evidence in this

regard includes: firstly, FMT effectively replicates the protective effects of aerobic exercise against weight gain, lipid accumulation and AS plaque formation in high-fat diet-induced AS mice; secondly, it ameliorates high-fat diet-induced dyslipidemia and reduces LPS levels; thirdly, it significantly enhances GM diversity, richness and abundance of beneficial bacteria (Fig 1b). Thus, our results suggest that FMT can serve as a viable approach to achieve the beneficial effects associated with aerobic exercise.

Increasing evidence suggests that HFD-induced GM imbalance [32] is one of the key pathogenic factors of AS [33]. By altering GM composition, FMT effectively improves AS lesions in susceptible mice, indicating its potential for remodeling the AS phenotype [26], which aligns with the outcomes of this study. Furthermore, previous clinical experiments have shown significant differences in GM between the aerobic exercise group and the AS group. Therefore, the impact of aerobic exercise on reshaping GM has been studied using a HFD-induced obese animal model. Building upon these findings, our study combines FMT with aerobic exercise and achieves promising results. The study differs from previous research, which mainly focused on FMT from healthy donors. In contrast, this study used aerobic exercise as the FMT intervention, resulting in a more significant improvement in AS.

HFD-induced GM imbalance is a significant contributor to dyslipidemia and inflammatory response [34], indicating its potential role in the development of AS. Obese individuals exhibit distinct differences in their GM compared to healthy individuals [35], and transferring of HFD-induced AS mouse GM to germ-free mice can induce dysbiosis, exacerbating perturbations in the GM composition and promoting blood lipid abnormalities, inflammatory responses, and AS phenotypes [7]. The study findings showed a significant improvement in blood lipid abnormalities among Group F participants. Additionally, there was a notable decrease in dyslipidemia-associated genera like *Lachnospiraceae_NK4A136*, *Dubosiella*, and *norank_f_Erysipelotrichaceae*. The GM metabolite LPS serves as a classical indicator of the status of the intestinal system and GM functionality. Increased levels of LPS are accompanied by alterations in intestinal permeability, leading to the occurrence of metabolic endotoxemia [36], which promotes inflammation and deposition of fat [37, 38]. This further elucidates the exacerbation of blood lipid abnormalities and inflammatory reactions in the AS group. Correlation analysis revealed a positive association between HFD-induced AS progression and LPS, while aerobic exercise and FMT intervention demonstrated an ameliorative effect on LPS elevation. Additionally, studies conducted by Lakio et al. [39] have demonstrated that LPS accelerates AS through the induction of foam cell formation and accumulation of LDL-C. Our research findings align with these discoveries. The findings of this study are in line with the aforementioned results. Moreover, it was observed that HFD consumption leads to elevated levels of LPS and bacteria associated with atherosclerosis, such as *Clostridium_sensu_stricto_1*, *Ileibacterium*, *Bifidobacterium*, *Dubosiella*, and *Faecalibaculum*; while there is a decrease in the abundance of *Prevotellaceae_UCG-001*, *Prevotellaceae_NK3B31_group*, and *Candidatus_Saccharimonas*. In contrast, different outcomes were observed in groups A and F due to the enhancement of beneficial bacteria and strengthening of intestinal barrier integrity caused by aerobic exercise intervention, thereby preventing the movement of LPS into the bloodstream, and FMT effectively migrated this result.

Exercise improves obesity and inflammation by remodeling GM to alleviate HFD-induced AS [40]. In the Bar graphs and LDA analysis, a significant increase in the relative abundance of specific bacterial genera associated with improved AS development was observed in Group A compared to the AS group. These genera include *Lachnospiraceae_NK4A136_group*, *Prevotellaceae_UCG-001*, and *Bacillus*. Since AS is characterized by chronic inflammation, anti-inflammatory mechanisms represent a primary therapeutic strategy. The surfactants produced by Bacillus can stimulate the production of various anti-inflammatory factors, including

interleukin 10 (IL-10), transforming growth factor-beta (TGF-β), and IgA in the host's intestinal tract [41], while also exhibiting a negative correlation with genes associated with fat accumulation, namely LARP4, ASB16, and THBS3 [42]. Furthermore, our Spearman correlation analysis provided evidence for a significant negative association between Bacillus abundance and serum LPS concentration, indicating an attenuation of chronic inflammation. *Prevotellaceae_UCG-001* has been found to be negatively correlated with histamine production [43], which contributes to elevated levels of inflammatory cytokines such as IL-6 and adhesion molecules that accelerate AS progression [44]. Therefore, the aerobic exercise-induced induction of *Prevotellaceae_UCG-001* plays a pivotal role in exerting anti-inflammatory and anti-atherosclerotic effects. It is worth noting that bacteria capable of producing butyrate, such as *Lachnospiraceae_NK4A136_group* and *Prevotellaceae_UCG-001*, enhance intestinal barrier integrity by generating SCFAs [45, 46], reducing the leakage of LPS, improving inflammation and endothelial cell damage, thus achieving the effect of improving AS [47]. This result is consistent with the findings in groups A and F in this study.

FMT modulates the microbial population through donor transplantation, reshapes the GM balance, and effectively treats diseases associated with GM alterations. Whether transferring the GM of healthy mice into HFD-induced obese mice [48], or transplanting the GM of obesity-modified mice into germ-free mice [49], in all cases, the donor's GM phenotype was faithfully replicated, thus further elucidating the causal role of FMT in reshaping GM and improving disease outcomes. Consistent with previous studies, our research demonstrated that compared to the AS group, both groups A and F exhibited increased diversity and abundance of beneficial bacteria in their GM profiles while showing a decrease in Gram-negative bacteria and harmful bacterial populations. Notably, we observed an increase in beneficial bacteria such as *Lachnospiraceae NK4A136*, *Prevotellaceae_UCG-001*, and *Bacillus* along with a reduction in *Faecalibaculum* and *Clostridium_sensu_stricto_1*—two bacterial genera known to promote inflammation [50]. *Faecalibaculum*, a pro-inflammatory bacterium, is predominantly distributed in the C and AS groups. LDA analysis also confirmed its dominance in AS mice, exhibiting a negative correlation with SCFA levels and a positive correlation with mouse weight, white adipose tissue, and LDL-C values [51]. On the contrary, aerobic exercise and FMT intervention not only observed a decrease in Faecalibaculum abundance but also improved body weight, lipid accumulation, and LDL-C levels, which is consistent with previous research findings [52]. Studies have revealed a strong correlation between the abundance of this bacterium and several inflammation-related genes such as indoleamine 2,3-dioxygenase 1 (IDO1) [52], which has been shown to promote AS progression [53]. Moreover, *Clostridium_sensu_stricto_1* can induce glucose and lipid metabolism disorders while being significantly negatively associated with AA and butyric acid (BA), thereby compromising intestinal barrier integrity [54]. Compared to the AS group, F group exhibited a significant reduction in *Clostridium_sensu_stricto_1* abundance. Our Spearman correlation analysis confirms a significant negative correlation between *Clostridium_sensu_stricto_1* abundance, HDL, and a significant positive correlation with TC. These results support our earlier discussion on *Clostridium_sensu_stricto_1*.

GM imbalance is an independent risk factor for AS, and our study confirms the association between FMT and AS phenotype as well as GM composition, while also identifying specific bacterial groups including *Lachnospiraceae_NK4A136_group*, *Prevotellaceae_UCG-001*, and *Bacillus*. However, the underlying mechanism by which GM or these specific bacterial groups exert inhibitory effects on AS remains elusive. Moreover, considering that FMT is typically administered once or multiple times, long-term safety concerns arise in its therapeutic application. Future research should focus on elucidating the mechanisms of GM remodeling to suppress AS while rigorously evaluating the safety profile of FMT.

In conclusion, our study validates that HFD-induced GM imbalance contributes to lipid accumulation, dyslipidemia, chronic inflammation, and ultimately AS. Moreover, FMT from mice subjected to regular exercise effectively attenuates HFD-induced AS by modulating GM composition. Our findings propose FMT as a promising therapeutic strategy for AS in lieu of aerobic exercise.

## 5. Conclusion

In summary, our study demonstrated that aerobic exercise-based FMT effectively mitigated lipid accumulation, ameliorated dyslipidemia and chronic inflammatory response through GM remodeling and replication of the aerobic exercise phenotype, thereby impeding AS development. Aerobic exercise-based FMT presents a promising therapeutic strategy for AS.

## Supporting information

**S1 Table. The ACE index and Shannon index of the gut microbiota samples of each group of mice.**
(XLSX)

**S1 Fig. Pathological state of the aorta stained with H&E after 10 weeks of rearing (group A).**
(TIF)

**S2 Fig. Pathological state of the aorta stained with H&E after 10 weeks of rearing (group AS).**
(TIF)

**S3 Fig. Pathological state of the aorta stained with H&E after 10 weeks of rearing (group C).**
(TIF)

**S4 Fig. Pathological state of the aorta stained with H&E after 18 weeks of rearing (group A).**
(TIF)

**S5 Fig. Pathological state of the aorta stained with H&E after 18 weeks of rearing (group AS).**
(TIF)

**S6 Fig. Pathological state of the aorta stained with H&E after 18 weeks of rearing (group C).**
(TIF)

**S7 Fig. Pathological state of the aorta stained with H&E after 18 weeks of rearing (group F).**
(TIF)

**S8 Fig. White adipose sections stained with H&E (group A).**
(TIF)

**S9 Fig. White adipose sections stained with H&E (group AS).**
(TIF)

**S10 Fig. White adipose sections stained with H&E (group C).**
(TIF)

**S11 Fig. White adipose sections stained with H&E (group F).**
(TIF)

**S1 Graphical abstract.**
(PNG)

## Author Contributions

**Conceptualization:** Jie Men, Hao Li.

**Data curation:** Hao Li.

**Formal analysis:** Hao Li.

**Funding acquisition:** Jie Men.

**Investigation:** Hao Li, Chenglong Cui, Xuedi Ma, Zhengyang Yu.

**Methodology:** Jie Men, Hao Li.

**Project administration:** Jie Men, Hao Li, Penghong Liu, Xueyan Gong, Youhao Yao, Jieying Ren, Chengrui Zhao, Binyu Song, Kaijiang Yin, Wei Liu.

**Resources:** Jie Men, Hao Li, Chenglong Cui.

**Supervision:** Jie Men.

**Validation:** Jie Men, Hao Li, Penghong Liu, Xueyan Gong, Youhao Yao, Chengrui Zhao, Kaijiang Yin, Jianting Wu, Wei Liu.

**Visualization:** Hao Li.

**Writing – original draft:** Hao Li.

**Writing – review & editing:** Jie Men, Penghong Liu.

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
