## [Decision Letter · Decision Letter 0]

28 Aug 2024

PONE-D-24-29677Fecal bacteria transplantation replicates aerobic exercise to reshape the gut microbiota in mice to inhibit high-fat diet-induced atherosclerosisPLOS ONE

Dear Dr. Li,

Thank you for submitting your manuscript to PLOS ONE. After careful consideration, we feel that it has merit but does not fully meet PLOS ONE’s publication criteria as it currently stands. Therefore, we invite you to submit a revised version of the manuscript that addresses the points raised during the review process.

Major:

1. Authors are requested to reanalyze the statistical significance test for figure1e, figure 3b, c. Authors are suggested to prepare all the bar graphs with individual data points for better understanding. Authors are also necessary to check that data has been represented either mean ± standard error or mean ± standard deviation as they claimed mean ± standard error.

2. Authors have mentioned in the material and methods section that they used Student's t-test and one-way ANOVA; otherwise, the Kruskal-Wallis. Authors are recommended to mention the tests used in Figure 1 a, e; Figure 2 and Figure 3 b, c with a number of samples/repeats (n =) in figure legends.

Minor:

1. Authors are suggested to add the significance test result in Figure 1a as described in result 3.1 section.

2. Authors are recommended to add the lipids’ names and LPS above the graphs of Figure 2 for easy visualization for readers.

3. Authors are suggested to correct the Excel data (supplementary) for the Ace and Shannon index which is written oppositely.

4. Authors are recommended to correct the appropriate reference mentioned as [16] in the Introduction.

We look forward to receiving your revised manuscript.

Kind regards,

Udai P. Singh, Ph.D.

Section Editor

PLOS ONE

Journal requirements: 1. When submitting your revision, we need you to address these additional requirements. Please ensure that your manuscript meets PLOS ONE's style requirements, including those for file naming. The PLOS ONE style templates can be found at https://journals.plos.org/plosone/s/file?id=wjVg/PLOSOne_formatting_sample_main_body.pdf and https://journals.plos.org/plosone/s/file?id=ba62/PLOSOne_formatting_sample_title_authors_affiliations.pdf. 2. To comply with PLOS ONE submissions requirements, in your Methods section in the MS, please provide additional information regarding the experiments involving animals and ensure you have included details on (1) methods of sacrifice, (2) methods of anesthesia and/or analgesia, and (3) efforts to alleviate suffering. 3. Thank you for stating the following financial disclosure:  [This study was supported by the National Natural Science Foundation of China (82201691); Cultivation Key laboratory of Metabolic Cardiovascular Diseases Research (TSGJ001); the Science and Technology Innovation Program of the Shanxi Provincial Department of Education (2023L472); the Science and Technology Plan Project of Lvliang City (2023SHFZ48); and the Talent Introduction Startup Fund of Fenyang College, Shanxi Medical University (2022A01).].  Please state what role the funders took in the study.  If the funders had no role, please state: ""The funders had no role in study design, data collection and analysis, decision to publish, or preparation of the manuscript."" If this statement is not correct you must amend it as needed. Please include this amended Role of Funder statement in your cover letter; we will change the online submission form on your behalf. 4. We note that Figure(s) Supplementary A, A2, AS, C, C2, F, White adipose (A), (AS), (F) and (C)  in your submission contain copyrighted images. All PLOS content is published under the Creative Commons Attribution License (CC BY 4.0), which means that the manuscript, images, and Supporting Information files will be freely available online, and any third party is permitted to access, download, copy, distribute, and use these materials in any way, even commercially, with proper attribution. For more information, see our copyright guidelines: http://journals.plos.org/plosone/s/licenses-and-copyright. We require you to either (1) present written permission from the copyright holder to publish these figures specifically under the CC BY 4.0 license, or (2) remove the figures from your submission: A. You may seek permission from the original copyright holder of Figure(s) Supplementary A, A2, AS, C, C2, F, White adipose (A), (AS), (F) and (C)  to publish the content specifically under the CC BY 4.0 license.  We recommend that you contact the original copyright holder with the Content Permission Form (http://journals.plos.org/plosone/s/file?id=7c09/content-permission-form.pdf) and the following text:“I request permission for the open-access journal PLOS ONE to publish XXX under the Creative Commons Attribution License (CCAL) CC BY 4.0 (http://creativecommons.org/licenses/by/4.0/). Please be aware that this license allows unrestricted use and distribution, even commercially, by third parties. Please reply and provide explicit written permission to publish XXX under a CC BY license and complete the attached form.” Please upload the completed Content Permission Form or other proof of granted permissions as an ""Other"" file with your submission.  In the figure caption of the copyrighted figure, please include the following text: “Reprinted from [ref] under a CC BY license, with permission from [name of publisher], original copyright [original copyright year].” B. If you are unable to obtain permission from the original copyright holder to publish these figures under the CC BY 4.0 license or if the copyright holder’s requirements are incompatible with the CC BY 4.0 license, please either i) remove the figure or ii) supply a replacement figure that complies with the CC BY 4.0 license. Please check copyright information on all replacement figures and update the figure caption with source information. If applicable, please specify in the figure caption text when a figure is similar but not identical to the original image and is therefore for illustrative purposes only. 5. Please include captions for your Supporting Information files at the end of your manuscript, and update any in-text citations to match accordingly. Please see our Supporting Information guidelines for more information: http://journals.plos.org/plosone/s/supporting-information. 

Additional Editor Comments:

Major:

1. Authors are requested to reanalyze the statistical significance test for figure 1e, figure 3b, c. Authors are suggested to prepare all the bar graphs with individual data points for better understanding. Authors are also necessary to check that data has been represented either mean ± standard error or mean ± standard deviation as they claimed mean ± standard error.

2. Authors have mentioned in the material and methods section that they used Student's t-test and one-way ANOVA; otherwise, the Kruskal-Wallis. Authors are recommended to mention the tests used in figure 1 a, e; figure 2 and figure 3 b, c with number of samples/repeats (n =) in figure legends.

Minor:

1. Authors are suggested to add the significance test result in Figure 1a as described in result 3.1 section.

2. Authors are recommended to add the lipids’ names and LPS above the graphs of Figure 2 to easy visualization for readers.

3. Authors are suggested to correct the Excel data (supplementary) for Ace and Shannon index which is written oppositely.

4. Authors are recommended to correct the appropriate reference mentioned as [16] in the Introduction.

Reviewers' comments:

Reviewer's Responses to Questions

**Comments to the Author**

1. Is the manuscript technically sound, and do the data support the conclusions?

Reviewer #1: Yes

Reviewer #2: No

2. Has the statistical analysis been performed appropriately and rigorously? 

Reviewer #1: No

Reviewer #2: No

3. Have the authors made all data underlying the findings in their manuscript fully available?

Reviewer #1: Yes

Reviewer #2: Yes

4. Is the manuscript presented in an intelligible fashion and written in standard English?

Reviewer #1: Yes

Reviewer #2: Yes

5. Review Comments to the Author

Reviewer #1: The study designed to focus on aerobic exercise influenced fecal microbiota transplantation (FMT) to reestablish the gut microbiota in mice to inhibit high-fat diet-induced atherosclerosis. They claimed that this technique is an innovative therapeutic strategy for clinical treatment of atherosclerosis. The manuscript is well written and easy to understand. However, several points need to be improved. Please, see the comments below.

Major:

1. Authors are requested to reanalyze the statistical significance test for figure1e, figure 3b, c. Authors are suggested to prepare all the bar graphs with individual data points for better understanding. Authors are also necessary to check that data has been represented either mean ± standard error or mean ± standard deviation as they claimed mean ± standard error.

2. Authors have mentioned in material and methods section that they used Student's t-test and one-way ANOVA; otherwise, the Kruskal-Wallis. Authors are recommended to mention the tests used in figure 1 a, e; figure 2 and figure 3 b, c with number of samples/repeats (n =) in figure legends.

Minor:

1. Authors are suggested to add the significance test result in figure 1a as described result 3.1 section.

2. Authors are recommended to add the lipids’ name and LPS above the graphs of figure 2 to easy visualization for readers.

3. Authors are suggested to correct the excel data (supplementary) for Ace and Shannon index which is written oppositely.

4. Authors are recommended to correct the appropriate reference mentioned as [16] in Introduction.

Reviewer #2: The manuscript entitled “Fecal bacteria transplantation replicates aerobic exercise to reshape the gut microbiota in mice to inhibit high-fat diet-induced atherosclerosis” sounds very interesting topic, however reviewer feels it need to modify little bit before published online.

# Line 33, 209, 241, 303 and 343: Unknown figure legends with different front sizes. Need do necessary Changes.

# Manuscript is pasted twice that need to be corrected and kindly use sample spacing format throughout the manuscript.

# Author must include any prebiotics, probiotics, and any standard weight loss medicine to compare the results, without that there is no positive control. Without that it is not rational to evaluate the study

6. PLOS authors have the option to publish the peer review history of their article (what does this mean?). If published, this will include your full peer review and any attached files.

Reviewer #1: **Yes: **Mousumi Mandal

Reviewer #2: **Yes: **Amarinder Singh

---

## [Author Response · Author response to Decision Letter 0]

24 Oct 2024

Response to Reviewers

Dear Editorial Board and Reviewers of PLOS ONE, 

Hello! We sincerely appreciate your time in reviewing our manuscript and providing valuable feedback on our paper titled "Fecal bacteria transplantation replicates aerobic exercise to reshape the gut microbiota in mice to inhibit high-fat diet-induced atherosclerosis" (Manuscript Number: PONE-D-24-29677R1). Your insightful comments have greatly contributed to improving the current version, and we are truly grateful for the opportunity to revise our submission. The authors have carefully considered your suggestions and have provided detailed responses addressing the raised issues. We hope that the revised manuscript, with all modifications highlighted in red, meets your standards. It is our sincere hope that you will consider accepting this revised paper for publication in PLOS ONE. Please find our responses to your comments below:

Response to Reviewer 1

Comment #1: Authors are requested to reanalyze the statistical significance test for figure1e, figure 3b, c. Authors are suggested to prepare all the bar graphs with individual data points for better understanding. Authors are also necessary to check that data has been represented either mean ± standard error or mean ± standard deviation as they claimed mean ± standard error.

Response from the Authors: 

Based on your valuable guidance and suggestions, our team has conducted an in-depth and meticulous review of the key data analyses in the manuscript, specifically addressing the statistical significance tests for the data presented in Figures 1e, 3b, and 3c. To ensure the accuracy and scientific rigor of the research results, we not only rechecked all related statistical processes but also conducted multiple verifications for potentially affected data sections, including a thorough review of the information presented in Figure 2.

During this process, we utilized GraphPad Prism 9.5 software tool - an industry-leading software - to comprehensively process, analyze, and graphically represent all relevant data. Ultimately, we have decided to express our data using the "mean ± standard deviation" format in Section 2.8 (Statistical Analysis) of the manuscript. This approach not only demonstrates our dedication to scientific rigor but also improves clarity in presenting research findings for peer reviewers and readers' accessibility. We are confident that these enhancements will further elevate both quality and credibility of this manuscript while appreciating your professional feedback that contributed towards refining and improving our work.

Comment #2: Authors have mentioned in material and methods section that they used Student's t-test and one-way ANOVA; otherwise, the Kruskal-Wallis. Authors are recommended to mention the tests used in figure 1 a, e; figure 2 and figure 3 b, c with number of samples/repeats (n =) in figure legends.

Response from the Authors: 

We greatly appreciate your suggestion regarding the lack of detailed annotations of the specific statistical analysis methods in the experimental data figures. We sincerely apologize for this oversight in the initial draft and have taken immediate steps to correct and improve this aspect. After thorough research and careful consideration, we confirm the following: Figure 1a was analyzed using Student's t-test; Figure 2 was analyzed using one-way ANOVA; and Figures 1e, 3b, and 3c were analyzed using the Kruskal-Wallis rank sum test, which is more appropriate for non-normally distributed data. Additionally, to enhance transparency and readability, we have precisely listed the sample sizes for each group in their respective figure legends, and we have also clearly indicated the specific statistical analysis methods used under each figure. These adjustments aim to ensure that our research report is more rigorous and standardized while demonstrating our commitment to academic integrity and high-quality scientific outcomes. Once again, we thank you for your insightful suggestions that undoubtedly helped us move towards higher standards.

Comment #3: Authors are suggested to add the significance test result in figure 1a as described result 3.1 section.

Response from the Authors: 

We sincerely appreciate and value your insightful suggestions, which have prompted us to carefully optimize Figure 1a in order to enhance its clarity and readability. Specifically, we have now marked the time point at week 11 in the figure, clearly indicating that this marks the official start of fecal microbiota transplantation (FMT) intervention for the F group mice at the end of week 11. To further emphasize the statistical significance of the changing trend, we have added ** at week 12 based on detailed significance test results and *** between weeks 13 and 19 to reflect significant differences in body weight data between AS and F groups over time. These subtle yet meaningful adjustments not only enrich the information presented in the figure but also offer readers a more precise perspective to observe experiment progression and key findings. Once again, we appreciate your thoughtful guidance that has enlightened our path towards continuously refining our research and striving for the highest standards in every outcome.

Comment #4: Authors are recommended to add the lipids’ name and LPS above the graphs of figure 2 to easy visualization for readers.

Response from the Authors: 

We have labeled the corresponding lipid names or LPS above each legend in Figure 2 to improve its readability.

Comment #5: Authors are suggested to correct the excel data (supplementary) for Ace and Shannon index which is written oppositely.

Response from the Authors: 

We sincerely apologize for misplacing the Ace and Shannon index data in Excel, which has now been corrected in the revised version. We greatly appreciate your scrutiny, and we will be more vigilant to prevent such oversights in the future, ensuring the maintenance of our research quality.

Comment #6: Authors are recommended to correct the appropriate reference mentioned as [16] in Introduction.

Response from the Authors:

We sincerely appreciate the profound insights you have provided. After carefully reviewing the references, we have made necessary changes in the revised version of the manuscript. Specifically, in the Introduction section, we have updated the reference content in [16] to: “Considering the contraindications and risks of exercise, there are obstacles to promoting exercise therapy for AS among the elderly population [1]; Fecal microbiota transplantation (FMT) offers a novel approach for treating this condition.” Once again, we deeply appreciate your keen observation. Your feedback serves as an indispensable source of motivation for our continuous pursuit of excellence.

1. Aengevaeren VL, Mosterd A, Sharma S, Prakken NHJ, Möhlenkamp S, Thompson PD, et al. Exercise and Coronary Atherosclerosis: Observations, Explanations, Relevance, and Clinical Management. Circulation. 2020;141(16):1338-50. Epub 2020/04/21. doi: 10.1161/circulationaha.119.044467. PubMed PMID: 32310695; PubMed Central PMCID: PMCPMC7176353.

Response to Reviewer 2

Comment #1: Line 33, 209, 241, 303 and 343: Unknown figure legends with different front sizes. Need do necessary Changes.

Response from the Authors: 

We greatly appreciate your meticulous feedback. In accordance with PLOS ONE guidelines, we have completely removed the blanks in #33, 209, 241, 303, and 343 from the revised manuscript. We sincerely apologize and thank you for bringing this to our attention.

Comment #2: Manuscript is pasted twice that need to be corrected and kindly use sample spacing format throughout the manuscript.

Response from the Authors: 

Your insightful suggestions have enabled us to reflect and improve. I am pleased to inform you that we have thoroughly reviewed the revised manuscript in accordance with PLOS ONE's stylistic standards, making necessary adjustments to ensure full compliance with the journal's requirements. We are committed to presenting a more rigorous and professional version of our work. Your attention and guidance serve as the driving force behind our progress, for which we extend our deepest respect and gratitude.

Comment #3: Author must include any prebiotics, probiotics, and any standard weight loss medicine to compare the results, without that there is no positive control. Without that it is not rational to evaluate the study

Response from the Authors:

We deeply appreciate your professional insights and support for our research. Your suggestion to include a positive control group containing prebiotics, probiotics, and standardized weight management drugs has greatly enriched the scope of our study and reflects a forward-thinking approach. However, conducting such supplementary experiments at this stage presents significant technical and methodological challenges due to the following reasons:

The primary challenge arises from concerns over sample stability due to the long time span of the project. This research has been ongoing for two years, and although we have employed low-temperature storage (-80°C) to preserve the integrity of the mouse fecal samples, previous studies indicate that long-term freezing may lead to a decline in α-diversity, likely due to microbial DNA degradation[2]. Consequently, restarting the animal model would not only require replicating the initial conditions accurately to ensure consistency for the newly added control group but also addressing uncertainties posed by variations in probiotic product efficacy, individual responses, and other external variables which could complicate comparability of experimental results.

Additionally, although an increasing body of evidence emphasizes that dysbiosis of gut microbiota (GM) is a potential contributor to atherosclerosis (AS), our understanding of its exact pathophysiological mechanisms remains limited [3]. While current research is gradually uncovering the functional interactions between GM and the host, the specific bacterial strains that should be introduced and the minimum effective dose required for intervention have not yet been clearly established[4]. A review of past studies[5, 6] shows that although changes in the microbiome and metabolites following FMT in AS model mice have demonstrated potential benefits, further systematic investigation is needed to comprehensively understand how these microbial components regulate the progression of AS as well as their distinct biological activities. Without a clear understanding of these underlying mechanisms, expanding the scope of FMT treatment could lead to unpredictable and difficult-to-interpret results.

Once again, we express our deep gratitude for your valuable feedback, which has significantly enhanced our study. We are committed to continuing our focus on investigating the molecular mechanisms through which exercise-mediated improvements in GM contribute to inhibiting AS. Our aim is to identify key bacterial strains and incorporate your suggestions into a more rigorous and refined experimental design. We firmly believe that by adhering to a rigorous, evidence-based scientific approach, we can unravel the complexities of biological phenomena and ultimately uncover the underlying causes of diseases, thereby contributing to human health and well-being.

2. Carroll IM, Ringel-Kulka T, Siddle JP, Klaenhammer TR, Ringel Y. Characterization of the fecal microbiota using high-throughput sequencing reveals a stable microbial community during storage. PLoS One. 2012;7(10):e46953. Epub 2012/10/17. doi: 10.1371/journal.pone.0046953. PubMed PMID: 23071673; PubMed Central PMCID: PMCPMC3465312.

3. Kim ES, Yoon BH, Lee SM, Choi M, Kim EH, Lee BW, et al. Fecal microbiota transplantation ameliorates atherosclerosis in mice with C1q/TNF-related protein 9 genetic deficiency. Exp Mol Med. 2022;54(2):103-14. Epub 2022/02/05. doi: 10.1038/s12276-022-00728-w. PubMed PMID: 35115674; PubMed Central PMCID: PMCPMC8894390.

4. de Groot PF, Frissen MN, de Clercq NC, Nieuwdorp M. Fecal microbiota transplantation in metabolic syndrome: History, present and future. Gut Microbes. 2017;8(3):253-67. Epub 2017/06/14. doi: 10.1080/19490976.2017.1293224. PubMed PMID: 28609252; PubMed Central PMCID: PMCPMC5479392.

5. Weingarden AR, Vaughn BP. Intestinal microbiota, fecal microbiota transplantation, and inflammatory bowel disease. Gut Microbes. 2017;8(3):238-52. Epub 2017/06/14. doi: 10.1080/19490976.2017.1290757. PubMed PMID: 28609251; PubMed Central PMCID: PMCPMC5479396.

6. Parker A, Romano S, Ansorge R, Aboelnour A, Le Gall G, Savva GM, et al. Fecal microbiota transfer between young and aged mice reverses hallmarks of the aging gut, eye, and brain. Microbiome. 2022;10(1):68. Epub 2022/05/04. doi: 10.1186/s40168-022-01243-w. PubMed PMID: 35501923; PubMed Central PMCID: PMCPMC9063061.

We sincerely appreciate the valuable feedback provided by the esteemed editorial board and reviewers on our manuscript. We eagerly anticipate your response

Thank you and best regards!

Yours sincerely,

Jie Men

Corresponding author:

Name: Jie Men

E-mail: menjie2020@126.com

---

## [Decision Letter · Decision Letter 1]

15 Nov 2024

Fecal bacteria transplantation replicates aerobic exercise to reshape the gut microbiota in mice to inhibit high-fat diet-induced atherosclerosis

PONE-D-24-29677R1

Dear Dr. LI,

We’re pleased to inform you that your manuscript has been judged scientifically suitable for publication and will be formally accepted for publication once it meets all outstanding technical requirements.

Kind regards,

Udai P. Singh, Ph.D.

Section Editor

PLOS ONE

Additional Editor Comments (optional):

The author did a wonderful job addressing the reviewer's concerns.

Reviewers' comments:

Reviewer's Responses to Questions

**Comments to the Author**

1. If the authors have adequately addressed your comments raised in a previous round of review and you feel that this manuscript is now acceptable for publication, you may indicate that here to bypass the “Comments to the Author” section, enter your conflict of interest statement in the “Confidential to Editor” section, and submit your "Accept" recommendation.

Reviewer #1: All comments have been addressed

2. Is the manuscript technically sound, and do the data support the conclusions?

Reviewer #1: Yes

3. Has the statistical analysis been performed appropriately and rigorously? 

Reviewer #1: Yes

4. Have the authors made all data underlying the findings in their manuscript fully available?

Reviewer #1: Yes

5. Is the manuscript presented in an intelligible fashion and written in standard English?

Reviewer #1: Yes

6. Review Comments to the Author

Reviewer #1: The manuscript entitled “Fecal bacteria transplantation replicates aerobic exercise to reshape the gut microbiota in mice to inhibit high-fat diet-induced atherosclerosis” focused on aerobic exercise influenced fecal microbiota transplantation (FMT) to reestablish the gut microbiota in mice to inhibit high-fat diet-induced atherosclerosis. The MS has been written well and easy to understand for the readers. The authors critically evaluated the FMT from aerobic exercise group of mice ameliorates high-fat diet-induced atherosclerosis. Authors have addressed carefully all the comments and suggestions of the reviewers in the revised version of MS. They also address the technical limitations for adding positive control in current stage. The work is promising for future therapeutics for atherosclerosis. According to my opinion, this manuscript is now suitable to accept for the publication.

7. PLOS authors have the option to publish the peer review history of their article (what does this mean?). If published, this will include your full peer review and any attached files.

Reviewer #1: **Yes: **Mousumi Mandal

---

## [Editor Report · Acceptance letter]

29 Nov 2024

PONE-D-24-29677R1 

PLOS ONE

Dear Dr. Li, 

I'm pleased to inform you that your manuscript has been deemed suitable for publication in PLOS ONE. Congratulations! Your manuscript is now being handed over to our production team.

Kind regards, 

on behalf of

Dr. Udai P. Singh 

Section Editor

PLOS ONE